# Therapeutic Efficiency of Multiple Applications of Magnetic Hyperthermia Technique in Glioblastoma Using Aminosilane Coated Iron Oxide Nanoparticles: In Vitro and In Vivo Study

**DOI:** 10.3390/ijms21030958

**Published:** 2020-01-31

**Authors:** Gabriel N. A. Rego, Mariana P. Nucci, Javier B. Mamani, Fernando A. Oliveira, Luciana C. Marti, Igor S. Filgueiras, João M. Ferreira, Caroline C. Real, Daniele de Paula Faria, Paloma L. Espinha, Daianne M. C. Fantacini, Lucas E. B. Souza, Dimas T. Covas, Carlos A. Buchpiguel, Lionel F. Gamarra

**Affiliations:** 1Hospital Israelita Albert Einstein, Sao Paulo 05652-900, Brazil; gabriel.nery@einstein.br (G.N.A.R.); javierbm@einstein.br (J.B.M.); fernando.anselmo@einstein.br (F.A.O.); luciana.marti@einstein.br (L.C.M.); igor.filgueiras@usp.br (I.S.F.); joãomatiasferreirav@gmail.com (J.M.F.); palomaespinha10@gmail.com (P.L.E.); 2Laboratory of Magnetic resonance (LIM-44), Faculdade de Medicina, Universidade de Sao Paulo, Sao Paulo 01246-903, SP, Brazil; mariana.nucci@hc.fm.usp.br; 3Laboratory of Nuclear Medicine (LIM-43), Departamento de Radiologia e Oncologia, Faculdade de Medicina, Universidade de Sao Paulo, Sao Paulo 01246-903, SP, Brazil; caroline.real@hc.fm.usp.br (C.C.R.); daniele.faria@hc.fm.usp.br (D.d.P.F.); buch@usp.br (C.A.B.); 4Faculdade de Medicina de Ribeirao Preto, Universidade de Sao Paulo, Ribeirao Preto, Sao Paulo 4049-900, Brazil; daianne.carvalho@hemocentro.fmrp.usp.br (D.M.C.F.); lucasebsouza@usp.br (L.E.B.S.); dimas@fmrp.usp.br (D.T.C.)

**Keywords:** magnetic hyperthermia, glioblastoma, SPION, nanoparticle, aminosilane, bioluminescence, PET/CT, motor behavior

## Abstract

Magnetic hyperthermia (MHT) has been shown as a promising alternative therapy for glioblastoma (GBM) treatment. This study consists of three parts: The first part evaluates the heating potential of aminosilane-coated superparamagnetic iron oxide nanoparticles (SPIONa). The second and third parts comprise the evaluation of MHT multiple applications in GBM model, either in vitro or in vivo. The obtained heating curves of SPIONa (100 nm, +20 mV) and their specific absorption rates (SAR) stablished the best therapeutic conditions for frequencies (309 kHz and 557 kHz) and magnetic field (300 Gauss), which were stablished based on three in vitro MHT application in C6 GBM cell line. The bioluminescence (BLI) signal decayed in all applications and parameters tested and 309 kHz with 300 Gauss have shown to provide the best therapeutic effect. These parameters were also established for three MHT applications in vivo, in which the decay of BLI signal correlates with reduced tumor and also with decreased tumor glucose uptake assessed by positron emission tomography (PET) images. The behavior assessment showed a slight improvement after each MHT therapy, but after three applications the motor function displayed a relevant and progressive improvement until the latest evaluation. Thus, MHT multiple applications allowed an almost total regression of the GBM tumor in vivo. However, futher evaluations after the therapy acute phase are necessary to follow the evolution or tumor total regression. BLI, positron emission tomography (PET), and spontaneous locomotion evaluation techniques were effective in longitudinally monitoring the therapeutic effects of the MHT technique.

## 1. Introduction

Glioblastomas (GBM) comprise a group of primary or secondary brain malignant tumors [1,2], which are the most aggressive and frequent tumor (55.4% of gliomas) of the central nervous system, and were classified by the WHO as level IV [3]. The GBM conventional treatment consists of tumor surgical resection followed by concomitant administration of radiotherapy and chemotherapy with temozolomide (TMZ). The treatment in most cases is palliative, non-curative, and only provides a short increase in survival for patients, which usually does not exceed 24 months [4,5]. These limitations justify decades of commitment in preclinical researches, pursuing alternatives and more effective therapies for GBM treatment.

Among these therapies, there is the magneto hyperthermia therapy (MHT) that has been improved and studied in vitro and in vivo, alone or combined with the traditional therapeutic protocols. MHT is a procedure that releases heat locally in the tumor by mediators such as magnetic nanoparticles (MNP) by exposure to an alternating magnetic field (AMF) with appropriate frequency and magnetic field strength [6]. The most commonly used MNP in this technique is the superparamagnetic iron oxide nanoparticle (SPION). SPION belongs to materials classified as ferrimagnetic because they have a core with a crystal structure of hematite (Fe_2_O_3_), magnetite (Fe_3_O_4_), maghemite (γ-Fe_2_O_3_) or mixed ferrites [7,8]. In addition, this material possesses a magnetization called superparamagnetism, which depends on their physicochemical characteristics, such as diameter, morphology, crystal structure, among others. SPION are characterized by the formation of highly magnetic moments when exposed to AMF, which completely disappears when AMF is turned off [9,10].

Another important point is the SPION coating with organic or inorganic molecules such as dextran, chitosan, silica, starch, and polydimethylamine, which can alter SPION biodistribution, toxicity, clearance, surface charge, colloidal stability, cellular uptake, among other functions [11]. A SPION possible coating is the aminosilane, a material known to enhance protein and cell adhesion [9]. Aminosilane also display low toxicity and has been widely used in the construction of complex and multifunctional drug delivery systems [12]. Some important characteristics of SPION coated with aminosilane (SPIONa) for the MHT application is their higher uptake by cells [9], their previous implication in preclinical studies [13,14,15,16,17], and referenced use in several phase I and II clinical trials [18,19,20,21].

The greatest challenges for therapeutic techniques to achieve efficiency are based on parameters optimization, equipment configuration and delivery system including NPM type, physicochemical characteristics, dose, and number of applications that is needed for tumoral tissue eradication. Similarly occurs with the MHT technique, where studies describe comparative analysis in order to establish the best parameters of AMF, periodicity and MHT fractionation [14,15,22,23,24,25,26]. Preclinical studies report a variety of parameters depending on tumor type, tumor localization, magnetic nanomaterial type and miscellaneous AMF configurations [13,14,15,22,23,25,26]. These variations have lead to difficulties in finding comparative parameters for results normalization, which requires analysis of specific absorption rate (SAR) and calibration protocols to establish a precise metrological index [27]. Thus, there is no consensus in most GBM in vivo studies for the best condition for MHT application, nor the reason for performing this therapy in multiple applications [13,14,15,22,23,24,25,26,28,29,30]. Variation in the number of MHT applications, its configurations, and the period for tumor evaluation are important aspects that need to be further explored. An additional important aspect of studies using MHT therapy in GBM models is the tumor induction out of brain region, this aspect of differents microenvironment is problematic for outcomes interpretations [31], and impair a correct translational MHT application.

MHT therapy outcome evaluation has been reported by different approaches using conventional techniques as follows: trypan blue [19,32,33], MTT [19,34,35,36], tunnel assay [32], live/dead assay [37,38], and Western blot [32] that in most cases have limitations for longitudinal analysis. 

The therapeutic outcome evaluation by long periods is necessary due to tumor relapse cases. Imaging techniques are ideal for reducing the cost and animal numbers during the follow-up, and also providing dynamic structural, functional, and molecular information of the therapeutic process [39]. The imaging techniques most used are: bioluminescence (BLI) [40,41,42,43], near-infrared fluorescence (NIRF) [44], magnetic resonance imaging (MRI) [45,46], positron emission tomography (PET) [47,48,49], single photon emission computed tomography (SPECT) [50], magnetic particle imaging (MPI) [51], computed tomography (CT) [52], and behavioral assessment [53,54], which also has been indicate as tools for functional preclinical evaluations of longitudinal processes [40,41,42,43,47,48,49].

The bioluminescence technique has shown to be an excellent tool due to the best cost-effectivity for therapeutic outcomes evaluation in glioblastoma model [40,41,42,43], by having the advantage of high sensitivity with low background [55,56,57]. This technique involves the luciferin application to in vitro or in vivo models; luceferin reacts with luciferase enzyme, contained only in implanted tumor cells, this reaction is dependent of factors such as Mg^2+^, adenosinetriphosphate (ATP) and oxygen. The positive reaction offers linear correlation between luciferase concentration and photon emission, and can be detected and measured in real-time by the bioluminescence imaging (BLI) system [57].

BLI has some limitations regarding the two-dimensional (2D) nature of this imaging technique, which turns difficult the establishment of the brain depth in which the tumor is situated. In addition, BLI signal also depends on D-luciferin distribution, signal depth, and tissue absorption [44]. 

Another important technique for therapeutic follow up analysis in glioma models is the PET that display functional and biological information, such as metastatic lesion and tumor level staging of III and IV [49,58,59,60,61,62]. PET has high diagnostic, prognostic, and therapeutic evaluation value when integrating with CT or MRI informations [61,62] for drug metabolism and PET radiotracers as ^18^F-2-fluoro-2-deoxy-D-glucose (^18^F-FDG) which evaluates glucose metabolism. High-grade gliomas, such as GBM, are known to consume high glucose levels. This glucose consumption in humans is correlated with highly angiogenic tumors [47,63], there is data available on this correlation in preclinical studies [64]. 

Preclinical studies usually report therapeutic progress evaluation by behavior assessment for long time follow up in different physiopathological conditions, but this information in antitumor therapy evaluation is scarce. Few studies evaluated spontaneous locomotion activity as a great potential for tumor progress analysis and the effects of different therapeutic modalities in liver tumor [65] and GBM models [54]. Recently, our group has developed a study on behavioral patterns regarding the progression of intracerebral C6-induced GBM model and its effect on locomotor movements [53], this last study was used as a comparative parameter for the therapy approached in the present study.

Therefore, the present study has the objective to evaluate the therapeutic effect of in vivo MHT technique in combination with SPION applied multiple times in brain tumor models. The best SAR value was previously determined in vitro using different combinations of frequency and magnetic field.

## 2. Results

### 2.1. Evaluation of the SPIONa Specific Absortion Rate

The SPIONa (100 nm) heating curves (time versus temperature) acquired through magnetic field of 100, 200, and 300 Gauss combined with oscillation frequencies of 309, 364, 420, 557, and 874 kHz (Figure 1A–E) showed that none frequency combined with 100 Gauss were able to achieve the 43 °C (therapeutic temperature). However, all frequencies combined with 200 and 300 Gauss fastly achieved the therapeutic temperature in less than 2 min, showing an inverse relation between time and frequency to achieve the therapeutic temperature. The time for SPIONa reaches 43 °C after being submitted to each combination of the magnetic field and frequency is indicate in Table 1.

SPIONa SAR values were obtained from heating curves, as shown in Figure 1F. The mean and standard deviation values of SPIONa SAR (W/g) are shown in Table 2. ANOVA test showed significative difference in the comparison between the magnetic fields (*p* < 0.0.001), oscillation frequencies (*p* < 0.001), as well as in the interaction between magnetic fields and oscillation frequencies (*p* < 0.001)

Analyzing the different combinations of frequencies and magnetic field using the post hoc test (Table A1), we could verify that all frequencies oscillation tested combined with 100 Gauss magnetic field did not achieved significant differences in SAR value (*p* > 0.05).

The comparison of SAR value obtained with 309 and 364 kHz of frequencies combined with 200 or 300 Gauss of the magnetic field also did not exhibited significant differences (*p* > 0.05). Thus, only the SAR values obtained the combination between high frequencies (420, 557 and 874 kHz) with 200 and 300 Gauss of magnetic field displayed significant differences (*p* < 0.001).

The highest SAR value was obtained by the combination of 557 kHz oscillation frequency with 300 Gauss magnetic field and it was the parameter chosen for the in vitro studies. Considering the fact that high frequencies can cause undesirable electric currents [66], another parameter with high SAR but low-frequency values, such as a combination between 309 kHz and 300 Gauss, was also chosen. Thus, two combinations of AMF were included for in vitro MHT evaluation in multiple applications: 557 kHz, 300 Gauss, and 309 kHz, 300 Gauss.

### 2.2. In Vitro Study

#### 2.2.1. Bioluminescence Signal Kinetics of C6 Cell Line Transduced with Luciferase

To evaluate maximum intensity and sensitivity of BLI signal emitted by C6 cells transduced with luciferase was performed a kinetics analysis during 490 min. The maximum intensity of BLI signal was obtained at 55 min after D-luciferin application followed by intensity reduction of signal. This pattern was observed in all cellular concentrations tested, showing high signal amplitude for high cellular concentrations. Thus, samples with 10^6^, 10^5^, 5 × 10^5^, 7.5 × 10^4^, 5 × 10^4^, 2.5 × 10^4^ and 10^4^ C6 cells showed the respective maximum intensities (120.0 ± 4.5) × 10^12^, (59.4 ± 4.05) × 10^10^, (11.9 ± 1.26) × 10^10^, (99.2 ± 6.3) × 10^9^, (73.1 ± 7.0) × 10^9^, (35.7 ± 1.2) × 10^9^ and (11.7 ± 1.4) × 10^9^ photons/sec, as shown in Figure 2. Images acquired in few time points were placed above the BLI curve to show the signal intensity changes according to the scale (Figure 2). The inserted figure (Figure 2) displays the curve amplification for lower cellular concentrations, in order to improve the details observed for different BLI intensities.

#### 2.2.2. Evaluation of the SPIONa Internalization into C6 Cells

The SPIONa internalization into C6 cells was detected through Prussian blue staining and Nuclear Fast red staining. Figure 3 shows the micrographs of the SPIONa internalization into C6 cells: using 100 µgFe/mL of SPIONa (Figure 3B,D) and 200 µgFe/mL of SPIONa (Figure 3F,H) and their respective controls (Figure 3A,C,E,G). SPIONa internalization was observed for either concentration tested. The highest SPIONa concentration (200 µgFe/mL) displayed a more intense internalization as shown in Figure 3F,H, at 4× and 20× magnification, respectively.

#### 2.2.3. Evaluation of in Vitro MHT as a Function of Multiple Applications

SAR values analyzed in item 2.1 displayed two sets of parameters for in vitro MHT application [(309 kHz_300 Gauss) and (557 kHz_300 Gauss)]. The effect of MHT evaluated by BLI signal, after each therapeutic application (Figure 4A–C) showed similar behavior on cellular proliferation between experimental groups, in which the C6 cells were unlabeled with SPIONa [(C6); (C6-f1-B; C6-f2-B), as f1 = 309 kHz, f2 = 557 kHz and B = 300 Gauss). The same occurred in C6_SPIONa group, in which the C6 cells were labeled with SPIONa, but were not exposed to AMF, showing that cell proliferation occurs indenpedent of SPIONa presence. These groups did not show significant differences for BLI signal (*p* = 1). However, groups exposed to SPIONa and AMF [(C6-f1-B with SPIONa); (C6-f2-B with SPIONa)] showed a reduction in BLI signal after MHT applications for both set of parameters compared to control. (Figure 4A–D).

The BLI intenstities of each group were quantified, as shown in Table 3 and the results for statistic of repeated measuraments displayed significant differences for number of MHT applications (*p* < 0.001, time effect), different experimental groups (*p* < 0.001) and interection between groups and number of MHT applications (*p* < 0.001).

Analyzing the therapeutic group with AMF presence and SPIONa internalization, a significant reduction was observed in BLI signal intensity compared to their respective controls (*p* < 0.001, red color), as shown in Figure 5.

After one MHT application, the C6-f1-B group that used frequency of 309 kHz showed significant BLI intensity reduction, decreasing from (4.365 ± 0.276) × 10^9^ to (1.748 ± 0.112) × 10^9^ photons/s (*p* < 0.001, Appendix A–Table A2), comparing cells without SPIONa and with SPIONa, respectively.

The C6-f2-B group that used 557 kHz of frequency also showed a significant reduction of the BLI intensity from (4.367 ± 0.276) × 10^9^ to (8.730 ± 0.873) × 10^8^ photons/s (*p* < 0.001, Appendix A—Table A2), comparing cells without SPIONa and with SPIONa, respectively.

In addition, comparing the BLI intensities between C6-f1-B group and C6-f2-B group both with SPION, the highest value of frenquency (C6-f2-B group) showed a significant reduction of BLI intensity in comparison with C6-f1-B group that used low frenquecy values (*p* < 0.001, Appendix A–Table A2).

After two MHT applications (the second application occurred three days after the first MHT application), as shown in Figure 5B, the BLI intensities had similar pattern of reduction described for the first MHT applicaton. Thus, the C6-f1-B group (309 kHz of frequency) showed significant reduction of the BLI intensity decreasing from (5.237 ± 0.329) × 10^9^ to (1.309 ± 0.824) × 10^9^ photons/s (*p* < 0.001, Appendix A—Table A3), comparing cells without SPIONa and with SPIONa, respectively.

The C6-f2-B group that used 557 kHz of frequency also showed a significant reduction of BLI intensity decreasing from (5.237 ± 0.329) × 10^9^ to (6.546 ± 0.412) × 10^8^ photons/s (*p* < 0.001, Appendix A—Table A3), comparing cells without SPIONa and with SPIONa. In addition, comparing the BLI intensities between C6-f1-B group and C6-f2-B group both with SPIONa, the highest value of frenquency (C6-f2-B group) showed a significant reduction of BLI intensity compared to C6-f1-B group that used low frenquecy value (*p* = 0.047, Appendix A—Table A3).

After submitted to three MHT applications (the third application occurred three days after the second MHT application), the BLI intensity results displayed a different pattern from others MHT applications (Figure 5C). In this case, the BLI intensity had a superior decay after submitted to AMF with low frequency value (C6-f1-B group with SPIONa, with 309 kHz), comapred with the group that used the high frequency value (C6-f2-B group with SPIONa, with 557 kHz). Thus, the BLI signal intensity decreased from (6.241 ± 0.392) × 10^9^ to (4.364 ± 0.274) × 10^7^ photons/s in the C6-f1-B group and from (6.197 ± 0.390) × 10^9^ to (5.673 ± 0.357) × 10^8^ photons/s in the C6-f2-B group.

Consequently, the BLI intensities showed significant difference between C6-f1-B and C6-f2-B groups with SPIONa in the first MHT application (*p* < 0.001; Appendix A—Table A3), in the second MHT application (*p* = 0.047), but not in the third MHT application (*p* = 0.463; Appendix A—Table A4). The MHT therapeutic effect evaluated by BLI intentisty showed that the C6-f1-B with SPIONa group had a more effective therapeutic result than C6-f2-B with SPIONa group, but without significative difference.

Figure 5D display significant difference in C6-f2-B with SPIONa group between the first and second MHT application (*p* < 0.001), but not between the second and third MHT application (*p* = 0.066). However, in C6-f1-B with SPIONa group over time analysis of BLI intensity showed significant differents between all over time comparisons, the first with second and second with third MHT applications (*p* < 0.001). Due to the pattern of evolution occurred over time in the C6-f1-B with the SPIONa group and the low risk of the generation of parasitic electric currents with the low value of frequency, the combination of AMF and frequency to be applied in vivo MHT process was 309 kHz and 300 Gauss.

Thus, the in vitro MHT analysis in multiple applications allowed us to choose the more effective parameters of AMF to apply into in vivo analysis, considering the high SAR value, better efficacy for multiple therapy applications and using lower oscillation frequencies and magnetic field.

### 2.3. In Vivo Study

#### 2.3.1. Evaluation of Tumoral Growth by Histological Analysis

The volumetric and morphological analysis of C6 cells implanted in the motor cortex using robotic stereotaxic was performed by histology using H&E staining. In the sham group was observed only the effect of Dulbecco’s Modified Eagle’s Medium: nutrient mixture F-12 (DMEM/F12) implanted (Figure 6A,B). After seven days of tumor induction, in the left hemisphere was observed the tumor development (Figure 6C,D), showing the migratory tendency to callosum corpus, more clearly observed at the 14th day (Figure 6E,F), at the 21st day the tumor already compresses the corpus callosum and achieve the ventricle ipsilateral. The tumor evaluation at 28th day, display a big tumor mass compressing the ventricle and exceeding their limits for the other hemisphere, this tumor is inadequate for MHT multiple applications due to animal short survival of 18 days

#### 2.3.2. Evaluation of MHT Multiple Applications in Vivo

After in vitro MHT evaluation was possible to choose the following set of parameters AMF, 300 Gauss of the magnetic field and 309 kHz of frequency oscillation. The MHT multiple applications were performed at the 14th, 17th and 21st days after tumor induction for the first, second, and third MHT applications, respectively. BLI images of all experimental groups were obtained at 13th day as the baseline measurement before the first MHT application, at 22nd day, two days after multiples MHT applications as an acute measure of therapeutic effect of MHT and at 32nd day, 12 days after multiples MHT applications, as late measurement of therapeutic effect of MHT (Figure 7A,B).

Comparing the BLI signal intensities of each experimental group, the sham group did not show any BLI signal after D-luciferin application (Figure 7A—first line of images, and Figure 7B—black color bars). The tumor group that did not receive the MHT treatment, showed significant increase in the BLI signal intensity over time (*p* < 0.05, time effect; gray bars of Figure 7B), 68% and 274% of tumor growth, comparing the BLI intensity at 22nd and 32nd with the baseline (13th day).

The groups that received MHT therapy showed reduction of BLI signal proportional to the number of MHT applications, refleting in the tumoral size visualized in BLI signal at 22nd day (Figure 7A- B - blue, green and red bars). The animal’s tumor groups that received one, two and three applications showed reduction of BLI signal—29.7%, 61.4%, and 94.9% related to the reduction of tumor size in comparison to the baseline, with significant difference between groups and over time (*p* < 0.001). However, the decay of tumor signal was not constant, since on 32nd day of evaluation, groups with one and two MHT applications, showed the respective increase of 119.6% and 59.7% in the BLI signal compared to 22nd day of measurement, and significant differences between groups (*p* < 0.001).

The tumor group in which MHT therapy was applied three times showed at 32nd day absense of tumor relapse, mantaining the decay of BLI signal during late evaluation. The over time comparison of this group showed significant differences between 32nd day measures with the baseline (*p* < 0.001). Thus, three MHT applications revealed to be an effective therapy with almost complete tumoral mass elimination. These results were ilustrated by ^18^F-2-fluoro-2-deoxy-D-glucose (^18^F-FDG) PET images (Figure 7C), in which the glucose metabolism was analyzed before (baseline) and after the three MHT applications, comparing the sham and Tumor+3MHT group. In the sham group, the ^18^F-FDG uptake was constant, comparing the time points. Instead, the Tumor+3MHT group presented important decrease in glucose metabolism (^18^F-FDG uptake) compared to the baseline.

#### 2.3.3. Spontaneous Locomotion Evaluation in Multiple MHT Applications 

The spontaneus locomotor activity was analyzed in the experimental groups of MHT multiple applications using four behavior parameters: slow horizontal movement (S-MOV), fast horizontal movement (F-MOV), slow rearing (S-REA), and fast rearing (F-REA) at 0, 7, 14, 19, 24, and 32 days after tumor induction, as shown in Figure 8. 

Basal measurament for all groups did not show significant differences in any parameter of the behavioral evaluation (*p* > 0.05). At 7 and 14 days (Figure 8A–D), before the first MHT application the groups with tumor induction showed a significant reduction of movement frequency (*p* < 0.05), with exception in F-MOV parameter for the sham group (without a tumor) that show significant difference compared to other groups submitted to tumor induction (*p* < 0.001, Figure 8B) 

In the acute MHT evaluation (at 14, 16, and 19 days after tumor induction), the frequency of S-MOV, F-MOV, S-REA, and F-REA showed a soft improvement after MHT, but was not durable for the tumor groups that received only one or two MHT applications. The late MHT evaluation of these groups (at 24th and 32nd days after tumor induction), the frequency of all parameter of movement became regressed, showing a significant difference between groups (*p* < 0.001). However, the tumor group that received three MHT applications after 19 days of tumor induction had a significant improvement in all parameters, mainly in horizontal movements (S-MOV and F-MOV) compared to the others tumor groups that received one or two MHT applications (*p* < 0.001), keeping this pattern constant until 32nd day.

Thus, the behavior assessment through spontaneous locomotor activity showed a decay of movement frequency in all parameters analyzed for all groups. The frequency of S-REA decayed until 14 days and after occurred small variation for tumor group with MHT therapy and sham group. For other parameters, only the late MHT evaluation was more decisive for discrimination between groups.

## 3. Discussion

The present study demonstrated the efficacy of MHT multiple applications for glioblastoma tumor model based on structural and functional evaluation, reaching functional improvement until the latest stages of evaluation. Thus, for this success, many paramenters of MHT therapy were evaluated and had important impact in the achievement of these results.

Characteristics as size, shape, coating, colloidal stability, surface charge, and magnetic properties influence the SPION interaction with tumor tissue and consequently with the MHT process [67,68]. In a previous study developed by our group, we have developed data on stability and hydrodynamic diameter (D_H_) distribution of SPIONa dispersed in aqueous solution. These data were generated by dynamic light scattering and demonstrated stability during 24 h of evaluation, without forming agglomerations, mantaining the 100 nm of D_H_ [69]. Similar results were reported by other groups that compared SPIONa with uncoated SPION disperse in aqueous solution and their stability. These studies have also shown that SPIONa do not display agglomeration [70], in agreement with the SPION coating influence on their stability and heating features [10].

Another SPION characteristic used in the MHT technique is the high SAR that depend on size, shape, composition, magnetization, magnetic interaction, and concentration of SPION, also as the frequency and strength of magnetic field applied [71], it is necessary to avoid magnetic field with the high oscillation frequencies due to heating by eddy current [72]. Therefore, in our study, a set of frequencies and magnetic fields were evaluated for the obtainment of SPIONa heating curves and posterior SAR values as shown in Table 2, hanging from 3.789 W/g (100 Gauss; 309 kHz) to the maximum value of 320.07 W/g (300 Gauss; 557 kHz).

Similarly, SPIONa SAR value was reported by Yuan et al. [73], using an AMF with 9.12 kA/m (114 Gauss) and 250 kHz of oscillation frequency to determine the SAR value of 14.96 W/g. The variability of AMF parameters to obtained the SPIONa SAR values turns difficulty the comparison between studies, but the similarity of the SPIONa heating capacity can be performed using the intrinsic loss power (ILP = SAR/(fxB^2^)) which results in the SPION proprieties excluding the influence of AMF and oscillation frequency [74]. In our study, the ILP was the ~0.680 × 10^−8^ Wg^−1^Oe^−2^Hz^−1^, acceptable value when compared to ILP value defined by Yuan et al. [73] study, calculated in 0.46 × 10^−8^ Wg^−1^Oe^−2^Hz^−1^, considering the inhomogeneities of the magnetic field.

A SPION used in clinical applications [16], coated with dextran was submitted to AMF (11 kA/m (137 Gauss; 150 kHz) and showed SAR value of 286 W/g and correspondent ILP value of 0.100 × 10^−8^ Wg^−1^Oe^−2^Hz^−1^. Other SPION type with similar features and applications, but coated with starch, also was characterized and showed the respective values of SAR and ILP; 230 W/g and 1.4 × 10^−8^ Wg^−1^Oe^−2^Hz^−1^ [75]. However, the ILP value produced by computational simulations for this SPIONa was 1.02 × 10^−8^ Wg−1Oe-2Hz-1. Thus, it is possible based on the previuos data to state that SPION coated with dextran has no potential for MHT, instead SPION coated with starch has potential for MHT [73]. Analyzing these results, we can affirm that the SPION coated with dextran used in the Ribas et al. study [16], have no potential for MHT, compared to the SPION coated with starch used in the Ludwig et al. study [75], and futher comparing the ILP values of SPION starch coated with SPIONa we can conclude that both have equal potential for use in MHT application. The difference in ILP values is attributed to the non-homogeneous magnetic field generated by the coil and the coating material used in the SPION [73,75].

On the other hand, an important aspect of MHT process is the SPION internalization for the cells into tumoral tissue, the surface charge (zeta potential) of SPION has influence in the pathways of SPION internalization by tumoral cells [67] due to the negative character of the lipidic bilayer found in the cell membrane external surface [76]. The positive character of SPION surface contributes favorably for their internalization process due to electrostatic action, as seen in C6 cells [77,78,79,80]. In our study, the SPIONa zeta potential displayed positive character (+20 mV) which favored the nanoparticle internalization into C6 tumoral cells as observed in our results. The SPIONa internalization was greater according to higher concentrations values, without any evidence of toxicity. This type of strategy was reported in other studies that used SPION coated with polyethyleneimine (PEI) with a positive surface charge of +29.28 mV, promoting their internalization in macrophages cells as much as in tumoral cells compared to SPION with zeta potential of −0.52 mV [11]. In addition, a comparative study of SPION internalization and toxicity showed good internalization rate and low toxicity for aminosilane-coated SPION with positive surface charge [12].

After the nanoparticles heating power evaluation through SAR and SPIONa internalization into C6 cells, was performed the in vitro study with MHT multiple applications. The BLI signal evaluation after the first and second MHT applications showed that the C6 cells viability was affected proportionaly to the applied frequency value. So, the cellular viability using AMF with high frequency (557 kHz; 300 Gauss), was reduced to 20.02% after one MHT and 12.49% after two MHT applications, for AMF low frequency (309 kHz; 300 Gauss), the viability was reduced to 40.03% and 25.02% after one and two MHT application, respectively.

In the third MHT application, the viability was inversely proportional to the frequency applied; the low and high frequency of AMF used showed the respective cellular viability reduction of 0.7% and 9.15%. Thus, in the third MHT application, the low efficiency of the AMF high frequency can be connected to a possible intracellular mechanism of heat stress resistance, likely related to heat shock proteins, such as hsp70 [13].

Several studies using in vitro MHT demonstrated different approaches and results, such as a study using SPION coated with aminolisane (50 nm) submitted to AMF with 220 kHz and 139.2 ampere (4.0 × 4.3 cm diameter coil) for 2 h showed 50% efficiency 24 h after finished the therapy. In this study, the SPION zeta potential was −10 mV, a value that might not favor the SPION internalization by cells [81]. Another study that synthesized aminosilane-coated SPION, they were submitted to low frequency (1.16 uT and 350 kHz) and a single MHT application for 30 min, showing 30% of K7M2 cells viability by BLI 24 h after finished the therapy [34], and others studies that used single MHT therapy application displayed efficiency superior than 50% [32,35].

Another studies in vitro using U-87MG cells and employing multiple MHT applications, used SPION with 106.2 nm of diameter submitted to AMF (750 kHz; 200 Gauss), and four days of MHT applications (1 per day during 2 h) showing 50% efficiency after the first MHT application and 80% after three days indicating the importance of latency period for MHT therapeutic efficacy analysis [30]. The precaution adopted in our study of two days latency for MHT efficiency analysis, was very important, because during this latency period it is possible the occurrence of cell death by membrane permeabilization or rupture, increased reactive oxygen species or heat shock protein (Hsp) expression [32]. Others studies that used two MHT applications did not showed significant differences in efficiency between first and second MHT applications, their evaluations occured right after the last MHT application within 30 min between applications [28,29].

Fractionated use of radiotherapy (or multiple applications) improves overall survival in patients, however, even though it has been applied for decades, there is still no optimal dose fraction established, although evidences suggest that moderately hypofractionated radiotherapy causes a survival increasing from 12 to 16 months [82,83]. Similar to radiotherapy, there is still no optimal dose fraction established for MHT therapy, and further studies are necessary to establish the best AMF parameters, periodicity of application, SPION dose and fractionation of MHT application. Although the MHT is already being applied in GBM clinical trials phase I and II as adjunctive therapy, there is few reports since 2009 [18,20,21,84], from in vivo preclinical studies supporting MHT fractional doses applications [14,15,22,23,24,25,26]. A source of variability concerns to the site of GBM tumor induction and many of these models are not induce intracerebrally but subcutaneously in non-brain regions, which may difficult later translation of the technique [13,14,15,25]. Though, this is changing, and more recently, studies have been developing intracerebral models of GBM with various cell lines [22,23,26]. Our study is the first to evaluate the effects of multiple MHT applications using C6 cells in an intracerebral model of GMB.

MHT therapy with multiple applications in vivo has been reported in literature with several diferent times of applications such as: 2 [85], 3 [13,14,15], 4 [25], 7 [26], 15 [24], 24 [23], and 27 [22]. Studies for MHT with multiple applications in non-cerebral glioma model [14,15] used the 11^th^ day after T9 cell induction, to initiate the MHT using the magnetite cationic liposomes (MCL) submitted to 118 kHz and 30.6 kA/m (384 Oe) for 30 min per session, with 24 h of intervals between 3 applications. These three applications resulted in the respective tumor reduction of 20%, 60%, and 87.5–88.9%, and the animals survival was evaluated up to 30 days after applications [15].

Studies involving brain tumor models (ALTS1C1 cells) that used oleic acid-coated SPION submitted to four MHT applications for 12 min at 37 kHz and 2.5 kA/m, increased the average survival from 11 (control) to 31 days [25]. Another study that used SPION submitted to AMF with 200 kHz and 300 Gauss and SAR value of 57 W/g in 15 MHT applications showed therapeutic success after 28 days of tumor induction with initial tumor size of 3 mm^3^ [24]. These results differ from ours, which used SAR value of 169.279 W/g when submitted to 309 kHz and 300 Gauss of AMF, resulting in the tumor reduction after three applications of MHT and enhanced survival compared to the control group.

For the comparative analysis between studies, it is important to consider the tumor volume and the concentrations of SPION administered into tumor tissue [22,23,24]. There is evidence that the same MHT parameters applied in different tumor sizes showed diverse results, tumor size of 3 mm^3^ disappear completely, diferently from tumor size of more than 25 mm^3^ that showed tumor regression in varied proportions [24]. A study with three MHT application (SAR = 52 W/g) per week over 16 days, with session lasting 30 min, was also unable to completely destroy the tumor mass due to tumor size (100 mm^3^), not preventing tumor recurrence [22,23]. The same did not occur in tumors with size between 1.5–2 mm^3^, displaying the importance of tumor size evaluation for proper therapy planning, which was done in the present study, the application of MHT started with similar tumor volumes.

Longitudinal follow-up of both tumor growth and therapeutic process can be performed by the BLI technique. Our study used this technique to verify the treatment efficiency during the acute phase to verify the effectiveness of therapy and its relation to the number of MHT application and in the late phase after treatment aiming to verify the tumor recurrence. However, one and two MHT applications were not suffient to prevent tumor recurrence in the late phase of evaluation, 32 days after tumor induction, which did not occured when three MHT sessions were applied. This last approach maintained the therapeutic efficacy observed in the acute evaluation.

Some other studies also used BLI technique to evaluate the MHT therapy efficiency. Chen et al. [26], evaluated the MHT therapeutic effect at 7, 14, 21, and 28 days after tumor induction, starting the therapy four days after tumor induction and having it applied for seven consecutive days for 1 h using the AMF parameters of 1 Tesla and 20 Hz. This study showed that only 60% of animals displayed low BLI signal during acute phase and during the late phase, the 40% animal with no BLI signal sustained the same condition with no tumor recurrence. Alphandèry et al. [22] evaluated MHT therapy during 56 days, applying 27 MHT sections (202 kHz; 27 mT) evaluating the therapeutic efficacy one day after each session and until 150 days (late stage), 100% of animals had no BLI signal 68 days after tumor induction without tumor recurrence at the late stages of evaluation (150 days after tumor induction). 

In addition to BLI, another imaging technique used for longitudinal studies is PET, which analyzes the decreasing in glucose metabolism by tumor cell after therapy. This analysis is possible due to the decreased hexokinase II enzyme activity in adjacent tissues [86], by reducing ^18^F-FDG uptake. This radiotracer (half-life of 109.7 min) is the most widely used for detecting the glucose uptake by tumor cells [86,87]. The main objective of using PET analysis in GBM cases is to differentiate tumor recurrence from radiation necrosis [61]. There is a linear relationship in PET signals between GBM cell number and ^18^F-FDG uptake levels by these cells, that is significantly higher in small cell lung cancer with activity [88]. In small animals, ^18^F-FDG reaches tumor internalization plateau approximately after 45 min of administration [89,90]. It is possible that the decrease in glucose consumption evidenced in our study is related to decreased nutritional intake due to antiangiogenic effects. ^18^F-FDG has been clinically and preclinically tested as a marker to analyze antiangiogenic effects. One study showed the efficacy of ^18^F-FDG analysis correlating the overall survival of 11 gliomas patients with the antiangiogenic effect caused by drugs as bevacizumab [91]. Some preclinical studies are capable to detect tumor vascularization [64]. A limitation of our study was the lack of information about antiangiogenic mechanisms since we have not used specific epitopes markers such as HIF1, VEGF, PDGFRβ, among others.

At the same time to the imaging, technique analysis performed evaluations of animals spontaneous locomotion, and this approach is viable, sensitive, and useful for longitudinal evaluations. The motor behavior assessment for the tumor growth analysis and the MHT therapeutic effect was correlated to the BLI and PET results, mainly in the late phase of the MHT evaluation. The F-MOV, S-MOV, F-REA, and S-REA frequencies in tumoral control group showed continuous regression due to the motor impairment related to the tumor evolution, corroborating our group previous study that delineated impairment of motor pattern in GBM tumor model at 7, 14, 21, and 28 days after tumor induction using C6 cells [53]. The animals’ behavior promoted by MHT therapeutic was assessed after each MHT application leading to a discrete improvement of movement frequency, but this occurred mainly after the first and second MHT application and was not maintained constantly, only after the third MHT application the animals’ behavior improvement was evidenced. The lack of studies with this approach after brain tumor induction associated with MHT therapy did not allowed comparisons of our results in the acute phase and late phase, which have shown motor function improvement only for the group that received three MHT applications.

Therefore, approaching MHT therapeutic processes with multiple applications, using aminosilane-coated SPION allowed an almost total regression of the tumor, and further evaluations after therapy acute phase is necessary to follow the tumor evolution or total regression. BLI, PET, and spontaneous locomotion evaluation techniques were effective in monitoring longitudinally the therapeutic effects of MHT technique.

## 4. Materials and Methods 

### 4.1. Magnetic Nanoparticles

The magnetic nanoparticles used in this study were the superparamagnetic iron oxide nanoparticles coated with aminosilane (SPIONa) containing magnetite nucleus (Fe_3_O_4_), coupled with functional amine group (NH_2_) (fluidMAG-Amine, Chemicell, Berlin, Germany), with density of ~1.25 g/cm^3^, hydrodynamic diameter of 100 nm, zeta potential of +20 mV, and SPION number of 1.8 × 10^15^/g approximately.

### 4.2. Evaluation of the Heating Potential of SPIONa

SPIONa (5 mg/mL) samples dispersed in aqueous medium, allocated in thermally isolated recipient were submitted to MHT process in different combinations of magnetic fields (100, 200 and 300 Gauss) and oscillation frequency (309, 364, 420, 557, and 874 kHz) of the AMF, using the DM100 system (nB nanoScale Biomagnetics, Zaragoza, Spain) until achieving the therapeutic temperature of 43 °C in order to obtain heating curves as function of time (Δ*T*/Δ*t*). The samples temperature was monitored by optical fiber (3204, Luxtron Corp., Santa Ana, California, USA). Samples initial temperature was 19 °C, and measurements were performed in quintuplicates.

The specific absorption rate (*SAR*) values were calculated in accordance with the following equation:(1)$$SAR=\frac{m_{SPION}C_{SPION}+m_{L}C_{L}}{m_{SPION}}{\left(\frac{\Delta T}{\Delta t}\right)}_{max}$$
where $m_{SPION}$ is the mass of SPION [kg]; $C_{SPION}$ is the nanoparticle specific heat capacity [J/(kg.K)]; $m_{L}$ is the mass of liquid [kg], and $C_{L}$ is the medium-specific heat capacity [J/(kg.K)].

### 4.3. In Vitro Study

#### 4.3.1. C6 Cells Culture

C6 cells were cultivated in the Dulbecco’s Modified Eagle’s Medium: Nutrient Mixture F-12 (DMEM/F12) (GIBCO^®^ Invitrogen Corporation, California, USA) supplemented with 12% fetal bovine serum (FBS) (GIBCO^®^ Invitrogen Corporation, CA, USA) and 1% antibiotic-antimycotic solution (GIBCO^®^ Invitrogen Corporation, CA, USA). These cells were maintained incubated at 37°C and 5% CO_2_ in an incubator Forma^TM^ Series II (Thermo Fisher Scientific, Massachusetts, USA). When necessary these cells were tripsinized using 0.25% Trypsin-EDTA solution (GIBCO^®^ Invitrogen Corporation, CA, USA). Aiming to evaluate the therapeutic process using the BLI technique, the C6 cells were transduced with luciferase lentiviral vector using an already established protocol [53].

#### 4.3.2. Kinetics of Bioluminescent Signal of Luciferase Transduced C6 Cells

For the BLI signal evaluation (the BLI maximum intensity signal and temporal evaluation of BLI signal as a function of cell number), 100µL of D-luciferin (30 mg/mL) was added in the subsequent C6 cells concentrations: 10^4^; 2.5 × 10^4^; 5 × 10^4^; 7.5 × 10^4^; 10^5^; 5 × 10^5^, and 10^6^ cells/well. BLI signal acquisition was performed using IVIS^®^ Lumina LT Series III equipment (Xenogen Corp, PerkinElmer. California, USA) with the following parameters: expossure time of 2 ms, binning of 2 and f/stop of 4. The BLI images were processed using the Living Image Software version 4.3.1. (IVIS Imaging System) in radiation absolute units (photons/s) through the ROI of 1 cm^2^.

#### 4.3.3. Labeling C6 Cells with SPIONa and Internalization Imagining

Culture wells containing 10^5^ C6 cells were plated with 2 mL/well of DMEM/F12. After 24 h, in the culture wells was added a concentration of 100 and 200 μgFe/mL of SPIONa (in triplicate) and incubated for 12 h. After incubation, C6 cells were washed with Phosphate Buffered Saline (PBS) (GIBCO^®^ Invitrogen Corporation, California, USA) and fixed for 60 min with 500 μL of paraformaldehyde 4% (PFA 4%) (Sigma-Aldrich, Missouri, USA).

The internalization evaluation was performed using prussia blue staining with 500 uL (5% potassium ferrocyanide, Sigma-Aldrich, St Louis, MO, USA, and 5% hydrochloric acid, Merck, Darmstadt, Germany) for 15 min and washed once with deionized water. Then, the nuclear fast red staining was performed with 1% (0.02 g of nuclear fast red in 2 mL of deionized water) for 10 min for nuclei counterstaining, and then quickly washed once more and analyzed by optical microscopy image. C6 cells stained images were registered using brighfield microscopy Eclipse TI-Nikon (Nikon Corporation, Tokyo, Japan).

#### 4.3.4. In Vitro MHT

In vitro MHT process was performed in three stages and therapeutic efficiency was evaluated through the BLI technique as shown in the experimental design displayed in Figure 9A. Samples of 10^6^ C6 cells were labeled with SPIONa for 12 h followed by washing. These cells were trypsinized, suspended in 40 μL of DMEM/F12 with their respective controls groups and submitted to MHT process according to the heating planning (Figure 9B) using the DM100 system that apply AMF to keep constant temperature of 43 °C. According to the SAR values previously calculated were chosen two combinations of magnetic field and oscillation frequencies of AMF [(f1:309 kHz-B:300 Gauss); (f2:557 kHz-B:300 Gauss)] to apply in the MHT process. For MHT process evaluation was used six groups: (i) C6 group, cells unlabeled; (ii) C6-SPION group, cells labeled with SPIONa; (iii) C6-f1-B group, cells unlabeled submitted to AMF(f1-B); (iv) C6-SPION-f1-B group, cells labeled with SPIONa submitted to AMF(f1-B); (v) C6-f2-B group, cells unlabeled submitted to AMF (f2-B); and (vi) C6-SPION-f2-B group, cells labeled with SPIONa submitted to AMF(f2-B); as shown in Figure 9C. The MHT was applied for 30 min at 0, 3, and 6 days and the temperature monitored using an optical fiber (Luxtron 3204). The BLI evaluations were performed at 2, 5, and 8 days, after the addition of 10µl of luciferin/sample and posterior signal quantification in radiation absolute units (photons/s). The experiment was performed in triplicate per experimental group.

### 4.4. In Vivo Study

Forty-five male Wistar rats, weighing between 250 to 350 g were acclimated at the Surgical Experimentation and Training Center (CETEC) of the Instituto de Ensino e Pesquisa Albert Eisntein. These animals were exposed at 21 ± 2 °C with a 12 h light/dark cycle. Access to food and water was ad libitum during the experiment. This vivarium is accredited by the Association for the Assessment and Accreditation of Laboratory Animal Care International (AAALAC International). Our study was approved by the Ethics in Animal Research Committee of the Hospital Israelita Albert Einstein; number 3126-17.

#### 4.4.1. Glioblastoma Tumor Inducion in Animal Model

The animals were anesthetized with isoflurane (2–4%) by SomnoSuite^®^ system (Kent Scientific Corporation, Torrington, USA) and placed into the Neurostar Robot stereotaxic StereoDrive^®^ instrument (NeuroStar GmbH, Tübingen, Germany), which holds the skull with ear bars and a clamp system that tightens against the frontonasal bone and the palate. After making a skin inscision on the dorsal region of the skull, the system was calibrated using the follow reference points: bregma; lambda and 2 mm right and left of central point between bregma and lambda. The target injection site was defined by the following stereotaxic coordinates: 2.0 mm (bregma; rostral-caudal axis), 2.0 mm (medial-lateral axis), and 3.0 mm (ventral-dorsal axis) of Paxinos Atlas [92] and 10^6^ C6 cells suspended in 10µl of DMEM/F12 was injected in the 3,6 µL/min, using the Hamilton syringe of 10 µL (Hamilton Company, Nevada, USA). The bone recomposition was performed with Dencrilay^®^ (Dencril Produtos Odontológicos, Brazil) diluted in self-curing acrylic based polymethyl methacrylate (Classic Dental Products, Brazil) and posterior suture of the animal.

#### 4.4.2. Evaluation of Tumor Growth by Histological Analysis

The histological tumor growth evaluation was performed at 7 (*n* = 5), 14 (*n* = 5), 21 (*n* = 5), and 28 days (*n* = 5) after tumor induction, together with the controls without tumor induction (*n* = 5). The brain was extracted and fixed with 4% of Paraformaldehyde (PFA) for 48 h, then dehydrated in 30% of sucrose for 48 h and frozen at −80 °C. The histological sections were taken with 20 µm of thickness at −20 °C using the cryostat CM 1850^®^ (Leica Microsystems, Wetzlar, Germany) and placed in the gelatinized histological blades (Gelatin from porcine skin G2500, Sigma-Aldrich, Missouri, USA) for H&E staining (Sigma-Aldrich, Missouri USA). Blades were digitalized using ScanScope AT Turbo^®^ equipment (Leica Microsystems, Wetzlar, Germany).

#### 4.4.3. In Vivo Experimental Design of Magnetic Hyperthermia

For in vivo MHT study experimental design (Figure 10), five experimental groups were established: (i) No_Tumor group, animals submitted to craniectomy and administered 10 uL of saline; (ii) Tumor group, animals submitted to tumor induction without the MHT therapeutic application; (iii) Tumor+1MHT group, animals submitted to tumor induction and one section of MHT therapeutic at 14th day after induction; (iv) Tumor+2MHT group, animals submitted to tumor induction and two sections of the MHT therapeutic at 14th and 17th day after induction; and (v) Tumor+3MHT group, animals submitted to tumor induction and three sections of the MHT therapeutic at the 14th, 17th, and 20th day after induction. The BLI signal evaluation was performed one day before each MHT application, as well as after 2 and 12 days of the third MHT application, having a common period for all groups at 13, 22, and 32 days after tumor induction, as shown in Figure 10. In addition, the MHT therapeutic process was evaluated through glucose metabolism using PET/CT before and after all the MHT process at 13 and 22 days of tumor induction. The behavioral analysis was performed at 0, 7, 14, 16, 19, 24, and 32 days.

##### In Vivo MHT Therapeutic Process

The MHT therapeutic process was evaluated with one, two and three applications at 14, 17, and 20 days after glioblastoma tumor induction (Figure 10). At 13th day of tumor induction was performed the tumor volumetric analysis by BLI, and the next day 40 μL of SPIONa was administered in four coordinates around the center of the tumor mass (1 μL/min) using the robotic stereotaxic equipment (NeuroStar GmbH, Tübingen, Germany) 5 h before the first MHT application. The MHT applications were performed using the best parameters of AMF and oscillation frequency determined by the in vitro study using the DM1000 system. The therapeutic temperature of 43 °C was monitored by an optical fiber (Luxtron 3204), and kept constant for 30 min.

##### Therapeutic Process Evaluation by BLI

The MHT therapeutic efficiency was evaluated by BLI signal of tumor mass at 13 days after tumor induction (before MHT application, control measurement) and at 16, 19, and 22 days that correspond to 2 days after each MHT application. In addition, the BLI evaluation was also performed after 12 days of the third MHT application (at 32nd day) in all groups, allowing comparative analysis between groups and their respective controls (Figure 10). For BLI image acquisitions were administered 1500 µL of D-luciferin intraperitoneal (15 mg/mL) and the parameters and analysis of the BLI images were similar to the ones used in the in vitro study.

##### Evaluation of MHT Therapeutic Effect by ^18^F-FDG-PET

The MHT therapeutic effect was also evaluated by glucose metabolism analysis using the PET technique and administration of ^18^F-FDG in the animals. The PET images were performed before (at the 13th day) and after three MHT applications (at the 22nd day), as depicted in Figure 10, using a small animal equipment Triumph^®^ II preclinical scanner (Gamma Medica-Ideas, Nortridge, CA, USA). ^18^F-FDG (38–40 MBq) was injected in the penile vein and after 45 min, the animal was posionited with the brain in the center of the field of view. Image was acquired for 30 min and reconstructed using OSEM 3D algoritm, 20 interactions, and 4 subgroups. Images were fused to a MRI template available in the PMOD^®^ software.

##### Evaluation of MHT Therapeutic Effect by Spontaneous Locomotor Activity

The behavior assessment through the global spontaneous locomotor activity was performed using the Infrared (IR) Actimeter LE 8825 systems (Actitrack, Panlab Harvard Apparatus, Barcelona, Spain), which is basically composed by a 2 dimensional (X and Y axes) square frame, a frame support and a control unit. Each frame has 16 × 16 infrared beams for optimal subject detection. Each animal was posicioned in the center of an arena and during 5 min, they were analysed for spontaneous locomotor activities: frequency of horizontal movement, frequency of vertical moviment and frequency of rearing moviment in slow and fast speed considering 5 s the time limit of speed classification. The data was acquired at days 0, 7, 14, 16, 19, 24, and 32 after tumor induction and processed by SEDACOM v2.0. software (panlab, Barcelona, Spain).

### 4.5. Statistic Analysis

The experimental data were demonstrated by mean and standard deviation and analyzed by JASP (0.9.0.1) and Origin 9.1 software (OriginLab, Northampton, Massachusetts, USA). The data with normal distribution was analyzed with a parametric test and for non normal distribuition was used a non-parametric test. The comparison between two groups was performed using t-student test or Mann-Whitney test and for more than two group comparison or repeated measures was used ANOVA test or Kruskal-Wallis test. In ANOVA test was considered the Post Hoc analysis for more comprehension of the results, comparing the factors two by two. For significative results was considered *p*-value less than 0.05.

## Figures and Tables

**Figure 1 ijms-21-00958-f001:**
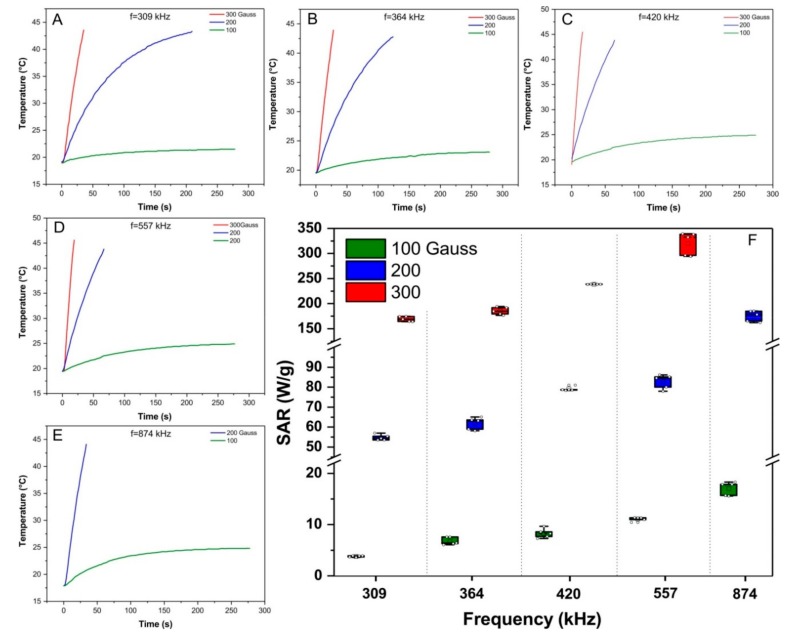
Evaluation of SPIONa heating potency exposed to the magnetic field of 100 (green line), 200 (blue line), and 300 Gauss (red line) combined with the oscillation frequencies of 309, 364, 420, 557, and 874 kHz. (**A**) Heating curves generated by SPIONa exposure submitted to alternating magnetic fields (AMF) with frequencies of 309 kHz; (**B**) 364 kHz; (**C**) 420 kHz); (**D**) 557 kHz; (**E**) 874 kHz. (**F**) Box plot graphic of SAR values calculated from the heating curves. SPIONa—aminosilane-coated superparamagnetic iron oxide nanoparticles.

**Figure 2 ijms-21-00958-f002:**
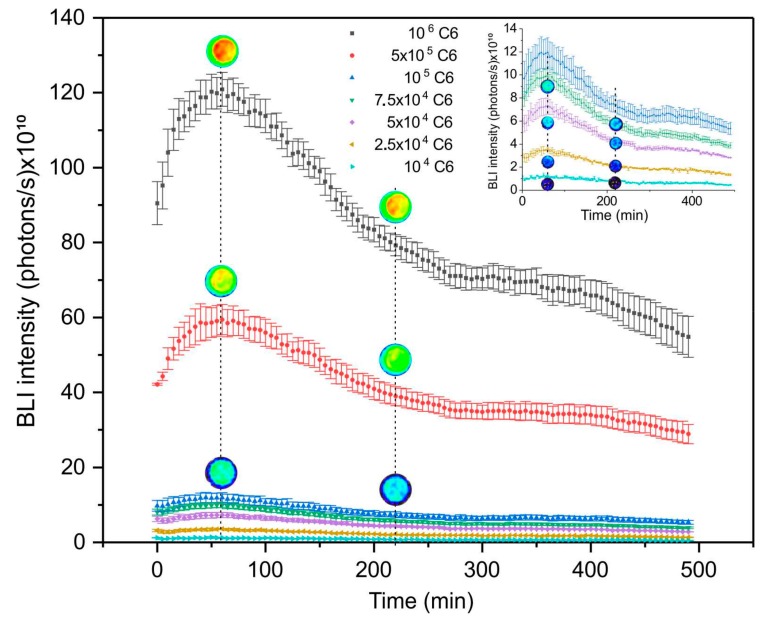
Bioluminescence kinetics signal as a function of C6 transduced with luciferase concentrations. The inset figure shows the curve amplification for lower cellular concentrations analysed for the variance between them.

**Figure 3 ijms-21-00958-f003:**
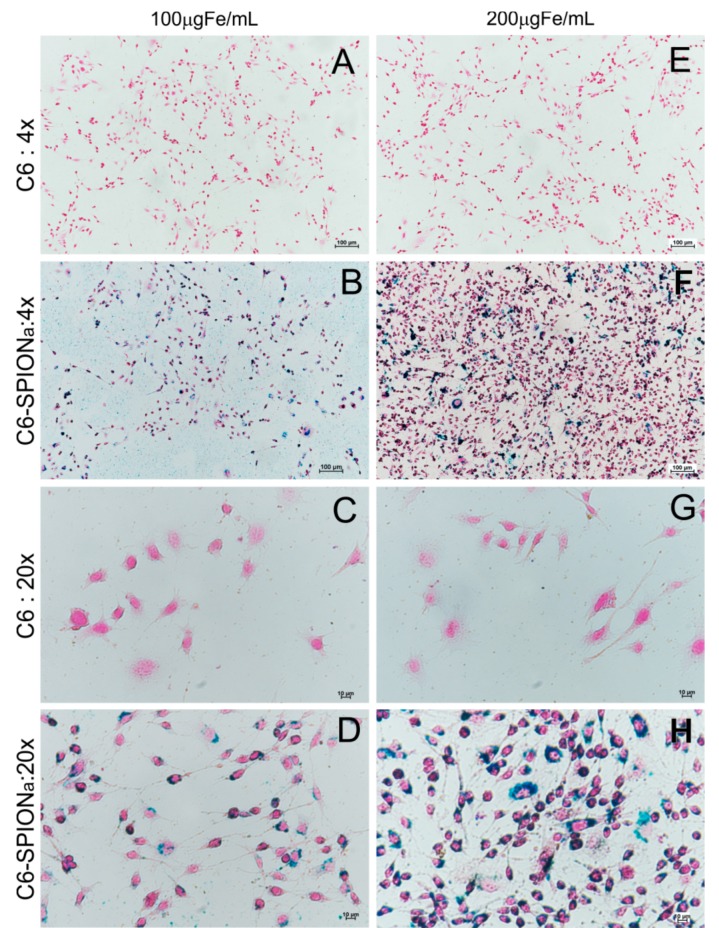
C6 cells labeled with SPIONa and stained with prussian blue and nuclear fast red. (**A**–**D**) Images of optical microscopy of the C6 cells labeled with 100 µgFe/mL of SPIONa concentration and respective controls and (**E**–**H**) 200 µgFe/mL of SPIONa and respective controls (4× and 20× magnification).

**Figure 4 ijms-21-00958-f004:**
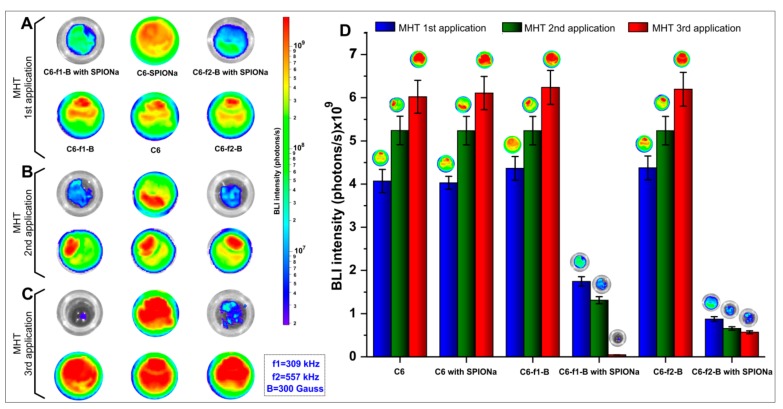
Evaluation of in vitro magnetic hyperthermia (MHT) therapeutic effect after multiple applications of bioluminescence imaging (BLI) signal: (**A**) BLI signal intensities after one MHT application, (**B**) BLI signal intensities after two MHT applications, (**C**) BLI signal intensities after three MHT applications, and (**D**) Histogram of BLI intensities in photons/s for each experimental group after one, two, and three MHT applications.

**Figure 5 ijms-21-00958-f005:**
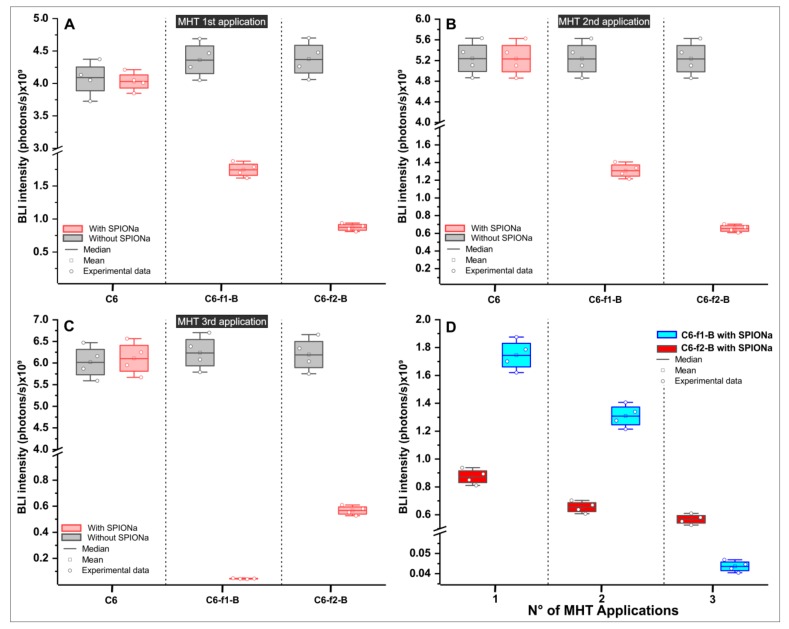
Comparison of BLI signal intensity between groups with and without SPIONa in the MHT multiple applications. The box plot shows the BLI intensity in the experimental groups (**A**) after one MHT application, (**B**) after two MHT applications, (**C**) after three MHT applications, and (**D**) the comparison of BLI intensities between the therapeutic groups, analyzing the two sets of AMF configuration: (f1-B e f2-B f1 = 309 kHZ; f2 = 557 kHz e B = 300 Gauss).

**Figure 6 ijms-21-00958-f006:**
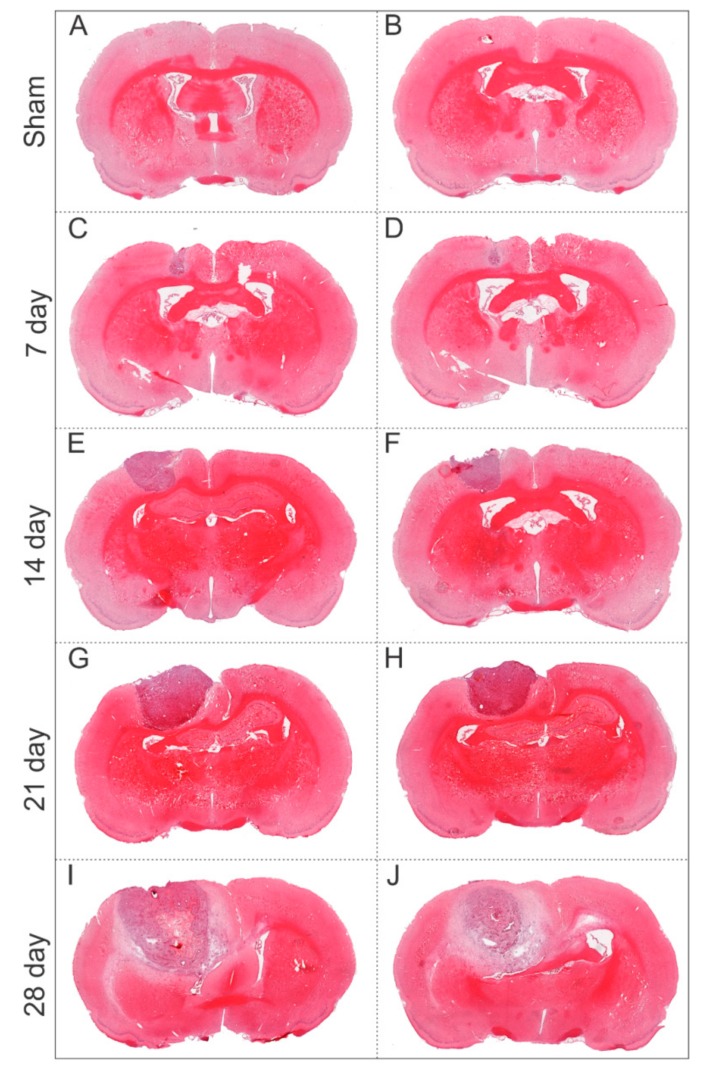
Evaluation of tumor growth by histology using H&E over 28 days after C6 cells tumor induction. (**A**,**B**) Sham group, brain without tumor induction, only medium injection. (**C**,**D**) Histological sections of tumoral tissue after seven days of tumor induction, (**E**,**F**) after 14 days, (**G**,**H**) after 21 days, and (**I**,**J**) after 28 days.

**Figure 7 ijms-21-00958-f007:**
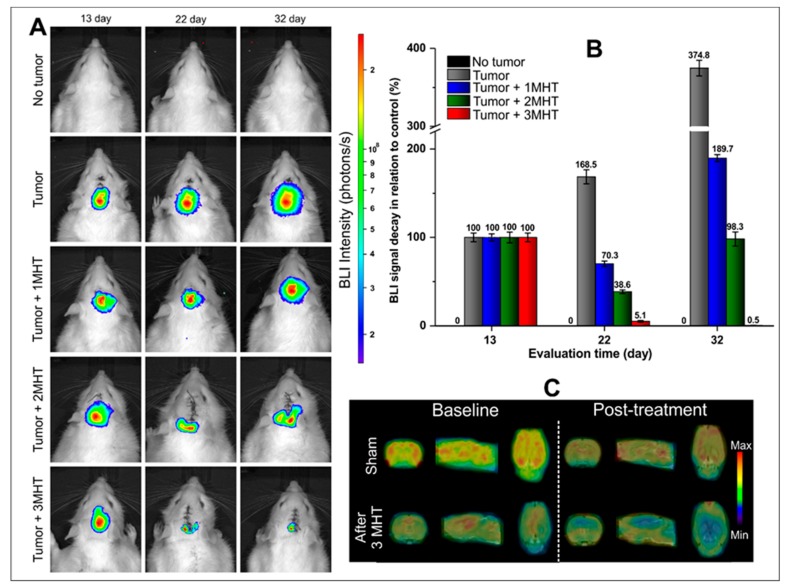
Evaluation of the MHT therapeutic process in multiple applications by BLI and ^18^F-FDG-PET. (**A**) BLI images of experimental groups at 13th (baseline), 22nd, and 32nd day after tumor induction, BLI signal changes according to the number of MHT applications. (**B**) Evaluation of BLI signals percentual after multiple applications of MHT therapeutic process. (**C**) Illustrative ^18^F-FDG-PET images fused to MRI templates of animals not submitted to tumoral induction (sham), animals submitted to tumor induction after three MHT applications, before (baseline) and post-treatment through ^18^F-FDG-PET.

**Figure 8 ijms-21-00958-f008:**
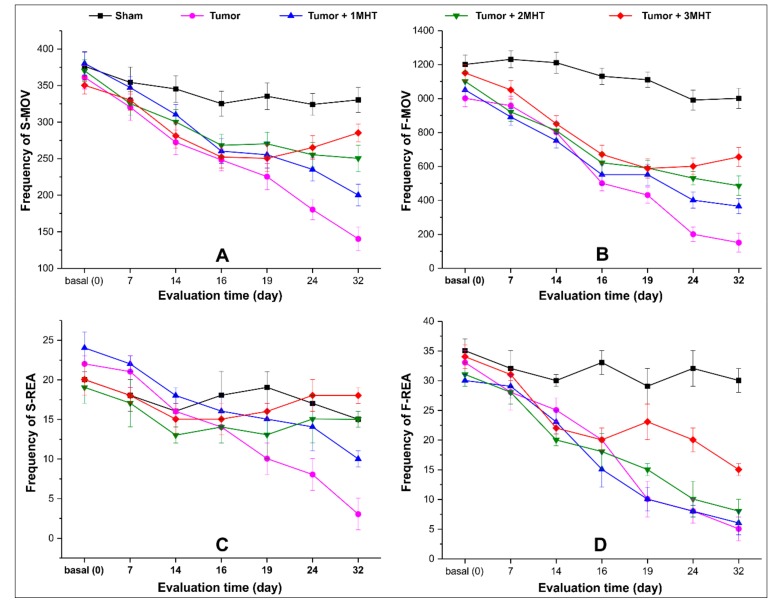
Spontaneus locomotor activity evaluation during multiples MHT applications. (**A**) Slow horizontal moviment (S-MOV), (**B**) fast horizontal moviment (F-MOV), (**C**) slow rearing (S-REA), and (**D**) fast rearing (F-REA), evaluted at 0, 7, 14, 16, 19, 24, and 32 day.

**Figure 9 ijms-21-00958-f009:**
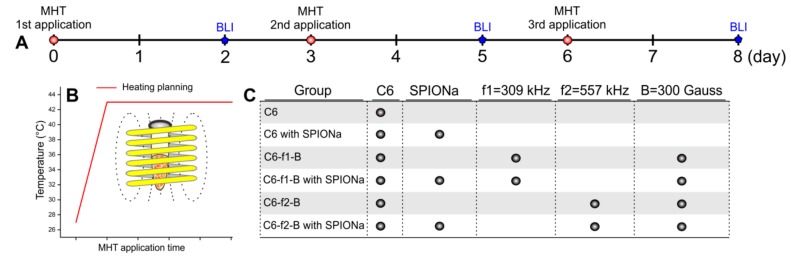
Experimental design of manifold applications of MHT in vitro. (**A**) Timeline with three MHT applications at 0, 3, and 6 days and their correspondent evaluation using BLI at 2, 5, and 8 days; (**B**) heating planning of the MHT process, keeping the constant temperature of 43 °C; (**C**) experimental groups and their respective controls submitted to MHT therapeutic process with two combinations of magnetic field and oscillation frequencies of the AMF [(f1:309 kHz-B:300 Gauss); (f2:557 kHz-B:300 Gauss)].

**Figure 10 ijms-21-00958-f010:**
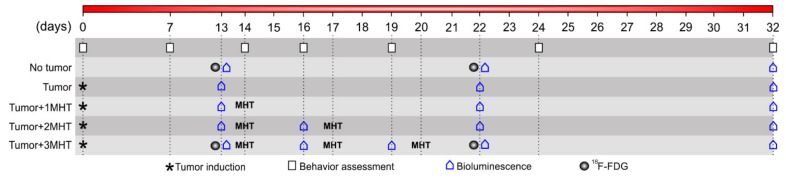
Experimental design, in vivo, for multiple MHT applications performed in 5 experimental groups: No_Tumor group, Tumor group, Tumor+1MHT group, Tumor+2MHT group, and Tumor+3MHT group were evaluated the therapeutic efficiency by BLI, ^18^F-FDG-PET/CT, and behavior.

**Table 1 ijms-21-00958-t001:** Heating time of SPIONa until achieving 43 °C (therapeutic temperature) as a function of magnetic field and oscillation frequency applied.

Magnetic Field (Gauss)	Frequeny (kHz)	Mean Time to Achieve of 43°C (s)	N
300	557	14 ± 2	5
300	420	15 ± 3	5
300	364	23 ± 4	5
300	309	26 ± 4	5
200	874	32 ± 5	5
200	557	57 ± 4	5
200	420	58 ± 4	5
200	364	104 ± 5	5
200	309	119 ± 7	5

kHz—kilohertz, SPIONa—superparamagnetic iron oxide nanoparticle coated with aminosilane.

**Table 2 ijms-21-00958-t002:** SPIONa SAR values for different combinations of magnetic field (100, 200, and 300 Gauss) and oscillation frequencie (309, 364, 420, 557, and 874 kHz).

Magnetic Field (Gauss)	Frequency (kHz)	SAR Mean (W/g)	SD (W/g)	N
100	309	3.789	0.137	5
	364	6.785	0.736	5
	420	8.295	0.920	5
	557	11.078	0.403	5
	874	17.045	1.306	5
200	309	54.757	1.460	5
	364	61.796	3.039	5
	420	79.117	1.101	5
	557	82.772	3.548	5
	874	175.112	10.897	5
300	309	169.297	5.097	5
	364	185.464	7.646	5
	420	238.775	0.350	5
	557	320.070	22.818	5

kHz—kilohertz, SAR—specific absortion rate, SD—standard deviation.

**Table 3 ijms-21-00958-t003:** BLI intensities values of samples submitted to in vitro MHT and their respective controls until three therapeutic applications.

MHT Applications	Groups	Mean (photons/s)	SD (photons/s)	*n*
One	C6	4.070 × 10^9^	2.643 × 10^8^	4
	C6 with SPIONa	4.030 × 10^9^	1.479 × 10^8^	4
	C6-f1-B	4.365 × 10^9^	2.763 × 10^8^	4
	C6-f1-B with SPIONa	1.748 × 10^9^	1.124 × 10^8^	4
	C6-f2-B	4.375 × 10^9^	2.763 × 10^8^	4
	C6-f2-B with SPIONa	8.730 × 10^8^	5.501 × 10^7^	4
Two	C6	5.242 × 10^9^	3.301 × 10^8^	4
	C6-SPIONa	5.237 × 10^9^	3.297 × 10^8^	4
	C6-f1-B	5.237 × 10^9^	3.297 × 10^8^	4
	C6-f1-B with SPIONa	1.309 × 10^9^	8.243 × 10^7^	4
	C6-f2-B	5.237 × 10^9^	3.297 × 10^8^	4
	C6-f2-B with SPIONa	6.546 × 10^8^	4.121 × 10^7^	4
Three	C6	6.022 × 10^9^	3.792 × 10^8^	4
	C6-SPIONa	6.110 × 10^9^	3.847 × 10^8^	4
	C6-f1-B	6.241 × 10^9^	3.929 × 10^8^	4
	C6-f1-B with SPIONa	4.364 × 10^7^	2.748 × 10^6^	4
	C6-f2-B	6.197 × 10^9^	3.902 × 10^8^	4
	C6-f2-B with SPIONa	5.673 × 10^8^	3.57 × 10^7^	4

MHT—Magnetic hyperthermia, SD—standard deviation.

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
