# Peer review of "Therapeutic Efficiency of Multiple Applications of Magnetic Hyperthermia Technique in Glioblastoma Using Aminosilane Coated Iron Oxide Nanoparticles: In Vitro and In Vivo Study"

_ijms, 2020, doi:10.3390/ijms21030958_

Round 1

Reviewer 1 Report

It a well written original paper evaluating the therapeutic effect in glioblastoma of in vivo magnetic hyperthermia technique applied multiple times in combination with multimodal nanoparticles exposed to alternating magnetic field standardized for best specific absorption rate value for outcomes in brain tumor models.

No further comments.

Author Response

Reviewer #1

It a well written original paper evaluating the therapeutic effect in glioblastoma of in vivo magnetic hyperthermia technique applied multiple times in combination with multimodal nanoparticles exposed to alternating magnetic field standardized for best specific absorption rate value for outcomes in brain tumor models.

No further comments.

English language and style

( ) Extensive editing of English language and style required 
( ) Moderate English changes required 
(x) English language and style are fine/minor spell check required 
( ) I don't feel qualified to judge about the English language and style 

Answer: Thank you for your observation. A native English speaker carried out an English language review and the manuscript was carefully corrected.

Reviewer 2 Report

Dear authors,

I recommend this manuscript for publication after minor revision. Pease edit some english phrases.

If you have collected the rat brains after the final experiments, you could H+E stain the tissue to show the reduction in tumor volume by another method.

Author Response

Reviewer #2

I recommend this manuscript for publication after minor revision. Pease edit some english phrases.

If you have collected the rat brains after the final experiments, you could H+E stain the tissue to show the reduction in tumor volume by another method.

Answer: Unfortunately we had problems in the preservation of tissue biological material due to damage in the freezer -80 ° C, but to complement the study we performed a functional analysis through the 18F-FDG PET evaluation to evaluate glucose metabolism by cells. tumors after the MHT process. Therefore, we did not perform immunohistochemical analyzes, but we focused on evaluations that could be performed longitudinally by imaging and behavioral techniques, showing the functional and structural effects of tumor volume decrease after therapies in acute evaluation and also in late evaluation, to detect a possible tumor recurrence, being more sensitive in monitoring the behavior of the tumor. Therefore, due to the relevance of the results found, we are starting a new study that will approach more deeply the possible therapeutic effects of MHT in relation to angiogenic processes caused by immunohistochemical glioblastoma tumor, as well as by H&E, HIF to mark hypoxia processes, VEGF to observe the expression of proangiogenic factors, MMP 2 and 9 to verify degradation of extracellular matrix for blood vessel formation and GFAP to verify astrocytic reactivity of adjacent tumor tissues, and these observations will be analyzed after each MHT application. in multiple applications.

English language and style

( ) Extensive editing of English language and style required 
(x) Moderate English changes required 
( ) English language and style are fine/minor spell check required 
( ) I don't feel qualified to judge about the English language and style 

Answer: Thank you for your observation. A native English speaker carried out an English language review and the manuscript was carefully corrected.
